# Comparing the Visual Perception According to the Performance Using the Eye-Tracking Technology in High-Fidelity Simulation Settings

**DOI:** 10.3390/bs11030031

**Published:** 2021-03-05

**Authors:** Issam Tanoubi, Mathieu Tourangeau, Komi Sodoké, Roger Perron, Pierre Drolet, Marie-Ève Bélanger, Judy Morris, Caroline Ranger, Marie-Rose Paradis, Arnaud Robitaille, Mihai Georgescu

**Affiliations:** 1Department of Anesthesiology, Faculty of Medicine, Montreal, Centre d’Apprentissage des Attitudes et Habiletés Cliniques (CAAHC), Université de Montréal, Montréal, QC H1T 2M4, Canada; tourangeau.mathieu@gmail.com (M.T.); roger.perron@umontreal.ca (R.P.); pierre.drolet@umontreal.ca (P.D.); marie-eve.belanger.7@umontreal.ca (M.-È.B.); arnaud.robitaille@umontreal.ca (A.R.); lmihai_georgescu@hotmail.com (M.G.); 2Cognitive Computing Department, Faculty of Science, University of Quebec at Montreal, Montréal, QC H3C 3P8, Canada; elisodoke@gmail.com; 3Department of Emergency Medicine, Faculty of Medicine, Université de Montréal, Montréal, QC H1T 2M4, Canada; judy.morris.med@ssss.gouv.qc.ca (J.M.); caroline.ranger.med@ssss.gouv.qc.ca (C.R.); marie-rose.paradis.med@ssss.gouv.qc.ca (M.-R.P.)

**Keywords:** eye-tracking, performance, medical education

## Abstract

Introduction: We used eye-tracking technology to explore the visual perception of clinicians during a high-fidelity simulation scenario. We hypothesized that physicians who were able to successfully manage a critical situation would have a different visual focus compared to those who failed. Methods: A convenience sample of 18 first-year emergency medicine residents were enrolled voluntarily to participate in a high-fidelity scenario involving a patient in shock with a 3rd degree atrioventricular block. Their performance was rated as pass or fail and depended on the proper use of the pacing unit. Participants were wearing pre-calibrated eye-tracking glasses throughout the 9-min scenario and infrared (IR) markers installed in the simulator were used to define various Areas of Interest (AOI). Total View Duration (TVD) and Time to First Fixation (TFF) by the participants were recorded for each AOI and the results were used to produce heat maps. Results: Twelve residents succeeded while six failed the scenario. The TVD for the AOI containing the pacing unit was significantly shorter (median [quartile]) for those who succeeded compared to the ones who failed (42 [31–52] sec vs. 70 [61–90] sec, *p* = 0.0097). The TFF for the AOI containing the ECG and vital signs monitor was also shorter for the participants who succeeded than for those who failed (22 [6–28] sec vs. 30 [27–77] sec, *p* = 0.0182). Discussion: There seemed to be a connection between the gaze pattern of residents in a high-fidelity bradycardia simulation and their performance. The participants who succeeded looked at the monitor earlier (diagnosis). They also spent less time fixating the pacing unit, using it promptly to address the bradycardia. This study suggests that eye-tracking technology could be used to explore how visual perception, a key information-gathering element, is tied to decision-making and clinical performance.

## 1. Introduction

Visual perception is a complex psychological operation used to understand our environment [1]. It is linked to higher cognitive function and highly affects the decision-making processes. The eye-tracking technology aims to report on some key elements of visual perception. It is commonly used in many fields such as marketing, neuroscience and industrial engineering [2,3,4]. Such uses rely on the suggestion that an individual’s gaze pattern is influenced by many things, including its knowledge level or expertise regarding a particular situation [5]. Several studies have linked the visual gaze pattern to expertise in the medical field. Katsuru et al. used eye-tracking technology to discriminate the differences in eye gaze between medical students and experienced physicians [6]. Literature research performed by Van De Luecht et al. to examine the visual processes of radiologists during lung nodule detection concluded that the eye-tracking technology could be a useful tool to inform advancements in lung screening [7]. Schultz et al. assessed workload in an anesthesia induction scenario in high-fidelity simulation. They concluded that pupil size and heart rate were not good comparison factors between individuals. The results were not statistically significant, but there was an inverse correlation between fixation and workload [5]. Additionally, Capogna et al. established, using eye-tracking technology, the importance of the distribution of the leaders’ gaze across all members of their team and the establishment of direct eye contact [8]. The association between the eye gaze pattern and the performance was also tested in other research fields such as aviation. Di-Flumeri et al. assessed, in real-time, the vigilance level of an Air Traffic Controller by measuring the time-to-first fixation as an indicator of a high level of vigilance [9]. Consequently, there seems to be a difference in gaze depending on the workload [10], and the experience of individuals. This difference in visual perception according to expertise is now well established. It incorporates mechanisms that capture the practice-related increase in the selective use of information [11], which leads to faster and accurate diagnoses. It is also based on a global-initial-focal search pattern, followed by a detailed, focal search-to-find mode [12]. The expert decision-making processes lead him to exhibit the more visual fixations in his relevant environment areas [13]. It goes without saying that a precise and rapid cognitive process should lead to a better performance. However, the current literature does offer only little evidence linking visual perception to performance, and basically compares the performance of the expert to that of the novice while monitoring their visual perception [14,15]. To our knowledge, no research compared gaze pattern and performance in a clinical simulation in a prospective and hypothesis-based manner. This exploratory work aimed to examine the visual perception of clinicians using eye-tracking technology during a high-fidelity simulation scenario and to look for differences between participants who succeeded and those who failed, apart from their expertise, when managing the critically ill simulated patient. The rationale of our hypothesis was to show that, with similar expertise, the performance of a physician during his practice depends on his situational awareness and understanding of his clinical environment. We hypothesized that the gaze patterns of physicians who managed the scenario successfully would be different from those who failed. As part of situational awareness building, we hypothesized that the practitioner would spend more time looking at the elements of his clinical practice setting that are unknown to him, but which would be crucial for the care of the patient.

## 2. Methods

The protocol was registered on ClinicalTrials.gov under the identifier NCT03049098, 9 February 2017. The institutional review board approved the research (Comité d’Éthique de la Recherche en Santé de l’Université de Montréal CÉRES, December 2017, #16-082-CERES-D-1) and written informed consents were obtained from all participants. A convenience sample of 21 first-year residents in their first three weeks of training from the emergency medicine department were enrolled voluntarily to be part of this exploratory study. We excluded those for whom it was not their first residency or those who are already licensed physicians in another country. Participants had to go through a 9-min high-fidelity simulation scenario in which a patient presented unstable bradycardia that needed to be externally paced. Throughout the scenario, individuals were wearing Tobii Pro Glasses^®^, an eye-tracking system that aggregates gaze data (https://www.tobiipro.com/fr/produits/tobii-pro-glasses-2/) (accessed on 23 January 2018). Glasses were fit to each resident’s face and calibrated following the manufacturer’s instructions. Prescription lenses were kept for calibration and during the experiment. During the scenario, the participant was allowed to move around and take care of the simulated patient as if he was in real clinical practice. In a time-wise fashion, we performed 3 calibration attempts to reach at least 75% accuracy and tracking scores, as it is recommended by the manufacturer. To do so, the participant stood one meter away from a whiteboard (a square of 1-m side) on which four infrared sensors were hung. After wearing the glasses (over his prescription lenses if necessary), the participant had to follow, only with his eyes, without moving his head, a 5th central sensor that the investigator made move following a Z-shape from bottom to top of the board, then from right to left. No calibration was needed after the experiment. Once calibrated, the squared IR AOIs can become distorted with peripheral vision, but the distortions do not affect the recognition of fixation in that AOI. The study took place at the University of Montreal high-fidelity Simulation Centre from 14–18 July 2017.

### 2.1. Scenario

This study was done using the two-mannequin-model (TMM) that was developed to improve our ability to use simulation to assess the clinical competency of the residents and the attending staff to treat severe bradycardia with hemodynamic instability. A high-fidelity mannequin (METI’s Human Patient Simulator, HPS, CAE Healthcare, Montreal, Canada) (https://caehealthcare.com/patient-simulation/hps/) (accessed on 23 January, 2018) on which blood pressure, electrocardiogram, saturation, respiratory rhythm, heart rate, and the pacing was controlled provided all physical findings and vital signs except for the ECG signal while another mannequin (SimMan 3G: https://laerdal.com/ca/products/simulation-training/emergency-care-trauma/simman-3g/) (accessed on 23 January 2018), located in an adjacent room, generated the ECG signal seen by the participants, which is sent to the modified HPS using a custom-made cable [16].

Individual data were obtained from pretests and general explanations about the simulation were given to participants. The scenario consisted of an 80-year-old patient hospitalized for pneumonia. The ECG signal showed a 3rd-degree atrioventricular block, a heart rhythm disorder that causes an extreme slowdown in the heart rate and results in a hemodynamic shock. Based on a recent publication addressing this specific scenario [16] successful management was defined by the participant properly installing the cardiac pacing material and achieving electrical and mechanical pacing of pulse in 9 min or less. Based on Ahn et al. [17], in a recently published list of items, clinical competency in transcutaneous pacing requires recognizing when TCP is indicated, properly applying the multifunction electrode pads and connecting them to a functioning pacing unit, properly applying the pacemaker’s (PM) electrocardiography (ECG) leads to the patient, setting the appropriate pacing mode, pacing rate and verifying ventricular capture. All other treatments, such as atropine were ineffective or declined. Unless pacing was set correctly, the patient continued to have nausea and to be hypotensive. A debriefing followed every simulation. 

### 2.2. Data Collection

Infrared (IR) markers were installed in the simulator and used to frame predefined Area of Interest (AOI). Four IR markers needed to be placed squarely to produce an AOI and allow data extraction. Three AOIs were created (Figure 1). 

The first one featured the head of the patient and the second one, the ECG and vital signs monitor. The defibrillator/pacing unit, the key device needed for successful management of the bradycardia, was featured in the third AOI. Tobii Studio^®^ software (Tobii Technology, Inc. Reston, VA, United States) was used to edit and aggregate gaze data and link them to each AOI. Information from IR markers was also used to build heat maps. Data collected included Total View Duration (TVD) or the time, in seconds, spent by a participant looking at an AOI during the whole simulation and Time to First Fixation (TFF) defined as the interval needed by the resident to take his first look at the AOI. 

### 2.3. Heat Map and Gaze-Trajectories Analysis

Heatmaps relative to each AOI were extracted. A heat map uses different colors depending on the number and on the duration of fixations that participants compiled in certain areas of the screen [18]. The heat map is a function of the TVD and the number of fixations.

### 2.4. Statistical Methods

Our sample size was a convenient one. Data are expressed as median and 25–75% quartile. Due to the limited sample size and the non-normal distribution of the data (D’Agostino and Pearson normality test), the Mann-Whitney U test was used to search for differences in TFF and TVD between the success and fail groups. A *p*-value under 0.05 was considered significant. To assess, in retrospect, whether our sample size was adequate, using the t value and the degrees-of-freedom (df) of the unpaired t-test, we calculated the effect size (r) to characterize the magnitude of the difference between the groups. An r value of 0.2 was considered a “weak” effect, 0.5 “medium,” and 0.8 “strong” [19]. Statistical analysis was performed using Prism 9.0.1 for Mac OS X (GraphPad Software, La Jolla, CA, USA).

## 3. Results

Twenty-one residents were enrolled. They were chosen following their registration order. Amongst them, three (17%) had some previous experience managing unstable bradycardia and one (5.5%) had placed pacing pads in a clinical setting. However, before simulation, most participants felt confident (61%) that they could manage unstable bradycardia. 

Three of the 21 participants were excluded; two due to technical difficulties during recording and the other because of failed calibration (10% accuracy), leaving 18 residents whose data were analyzed. Twelve participants managed the scenario successfully while six failed to do so. The TVD spent looking at the pacing unit was shorter for the residents who managed the simulation successfully compared to the ones who failed (42 [31–52] sec vs. 70 [61–90] sec, *p* = 0.0097). The group who succeeded also showed a shorter TFF before looking at the monitor compared with those who failed (22 [6–28] sec vs. 30 [27–77] sec, *p* = 0.0182) (Figure 2 and Figure 3). The effect size r was respectively 0.44 and 0.45 for the TVD of the defibrillator and the TFF of the monitor. Although it was significant, this indicates a “medium” effect between the groups.

TFF and TVD of the AOI head of the patient were similar between the groups (TFF: 9 [4–17] sec vs. 4 [3–12 sec, *p* = 0.2908] and TVD: 17 [5–34] sec vs. 10 [4–26] sec, *p* = 0.6820). 

We made comparisons of the TVD between the three AOIs, among participants who passed the scenario and among participants who did not. The results showed a similar gaze distribution profile among AOIs regardless of the outcome of the scenario. All participants spent significantly more time looking at the defibrillator than at the patient’s head or the monitor. They also spent a similar amount of time looking at the monitor and the patient’s head (Table 1).

## 4. Discussion

### 4.1. Key Results

Results showed that the residents who managed the scenario successfully looked at the ECG monitor, the only place where bradycardia was shown, earlier than those who failed. Additionally, participants succeeded looked at the pacing unit for a shorter time than their colleagues who failed. To our knowledge, this is one of the few studies that collected gaze patterns in participants having roughly the same expertise and looked for a link with their performance. By cons, several studies found an association between expertise and visual perception. In radiology, Cooper et al. used eye-tracking during the showing of CT-scan slices displaying different pathologies to participants with various levels of medical training. They found differences in visual search patterns between novices and experienced radiologists [20]. Turgeon et al. compared search patterns of fourth-year dental students and licensed oral and maxillofacial radiologists that were shown normal and pathologic panoramic radiographic images. Although all participants gazed longer at normal images, experts observed the normal images for a longer time and the pathologic images for a shorter time than dental students did [21]. Wilson et al. used a laparoscopic simulator linked to eye-tracking technology to compare experienced and novice surgeons. In this experiment, participants needed to touch flashing balls with laparoscopic tools. Results showed that experts watched the tools for a shorter time and the ball for a longer time than novices did [22]. The eye-tracking technology allowed the distinctions in visual perception of the experts and novices [23,24,25,26]. Typically, novice clinicians tend to unnecessarily activate data-collection approach to validate their hypothesis before arriving at a final diagnosis while experts make appropriate assumptions more quickly based on past similar experiences and accurately perceive more precisely the relative importance of information such as case progression, vital signs, and symptoms [27]. Experts frequently exhibit ‘‘verification-driven visual perception’’; they will typically display wide and few saccades with less erratic eye movements. On the contrary, novices typically display narrow and more saccades with more erratic eye movements [28,29]. 

Research in the field of visual perceptions also suggests a relationship between workload increase and the decrease in fixation duration [30]. In the aviation field, Thomas et al. observed a different gaze pattern between pilots who succeeded during an off-normal flight and those who failed. Pilots who detected the problem scanned the cockpit dashboard more often, whereas the pilots who failed ignored information [31]. This is consistent with our research. Indeed, we noticed that participants who succeed the patient’s management stared at the monitor more quickly. The rapid fixation of the monitor could allow to discover the bradycardia and to initiate the management more promptly. De Rivecourt et al. showed that with increasing workload, trained pilots tend to decrease fixation duration [32]. Their results perfectly support our research, as we demonstrated that amongst untrained medical students unfamiliar with bradycardia, the ones who fixated the pacing unit longer were mostly unsuccessful.

There is also a difference in gaze patterns depending on the knowledge level of the participant [33,34] as demonstrated in radiology, nursing, and dental education [35,36,37]. Cooper et al. used eye-tracking technology in radiology where they showed CT-scan slices displaying different pathologies to participants at different levels of medical training. They have shown differences in visual search patterns mainly between novices and experienced radiologists [38]. Turgeon et al. compared search patterns of fourth-year dental students and certified oral and maxillofacial radiologists that were shown normal and pathologic panoramic radiography. All participants gazed longer at normal images [21]. Certified oral and maxillofacial radiologists observed the normal images for a longer time and the pathologic images for a shorter time than dental students. Wilson et al. [22] used a laparoscopic simulator linked to eye-tracking technology to compare experienced and novice surgeons. In this experiment, participants needed to touch flashing balls with laparoscopy instruments. Results showed that experts watched the tools for a shorter time and the ball for a longer time than novices. 

Combining wearable EEG and eye-tracking makes it possible to refine the analysis of the gaze and to take one more step to integrate information processing with visual perception [39,40,41,42]. 

### 4.2. Limitations

There are several limitations to this exploratory study. An obvious limitation is its small sample size. Due to the participants’ busy schedules and our concern about the confidentiality of the scenario, we could only enroll 21 participants of which 3 were excluded because of technical problems. Despite the limited number of participants, the calculation of the effect size showed a “medium” effect, which would suggest that the number of additional participants would not be major. Our observational study will undoubtedly, as a seed study, serve to calculate the sample size of other similar ones.

Naturally, the external validity of the results remains unknown. We can only speculate whether the findings would have been different had we opted for another scenario. Additionally, the AOIs were chosen because they contained elements that were deemed to be the most relevant to the situation such as the monitor [5,43] or the mannequin’s head [44]. Nonetheless, these were arbitrary decisions, and we cannot exclude that other visual elements could have been of interest to the participants. Ending the scenario after 9 minutes was also an arbitrary decision although previous work conducted at our center [16] suggests that allowing participants who fail this type of scenario more times is unlikely to bring more understanding. 

## 5. Conclusions

In a high-fidelity simulation scenario, there were significant differences between the environmental visual interests of the first-year residents who performed successfully and those who failed. This suggests that the eye-tracking technology could be used to look into the relationship between key situational awareness elements like visual perception and clinical performance. Other research methods such as the think-aloud protocol, the simulation with iterative discussions [45] or the debriefing on-demand need to be performed before concluding the relationship between clinical reasoning, visual perception, and clinical management.

## Figures and Tables

**Figure 1 behavsci-11-00031-f001:**
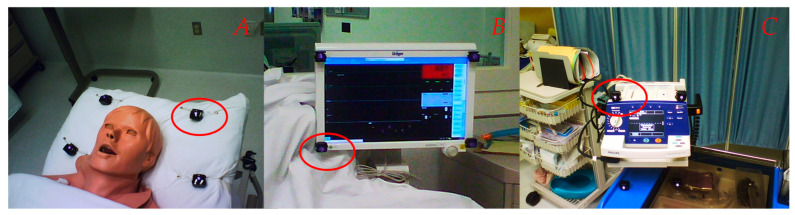
Infrared markers (red oval) framing three different Areas of interest (AOI): (**A**) Head of the patient, (**B**) Monitor, (**C**) Defibrillator/pacing unit.

**Figure 2 behavsci-11-00031-f002:**
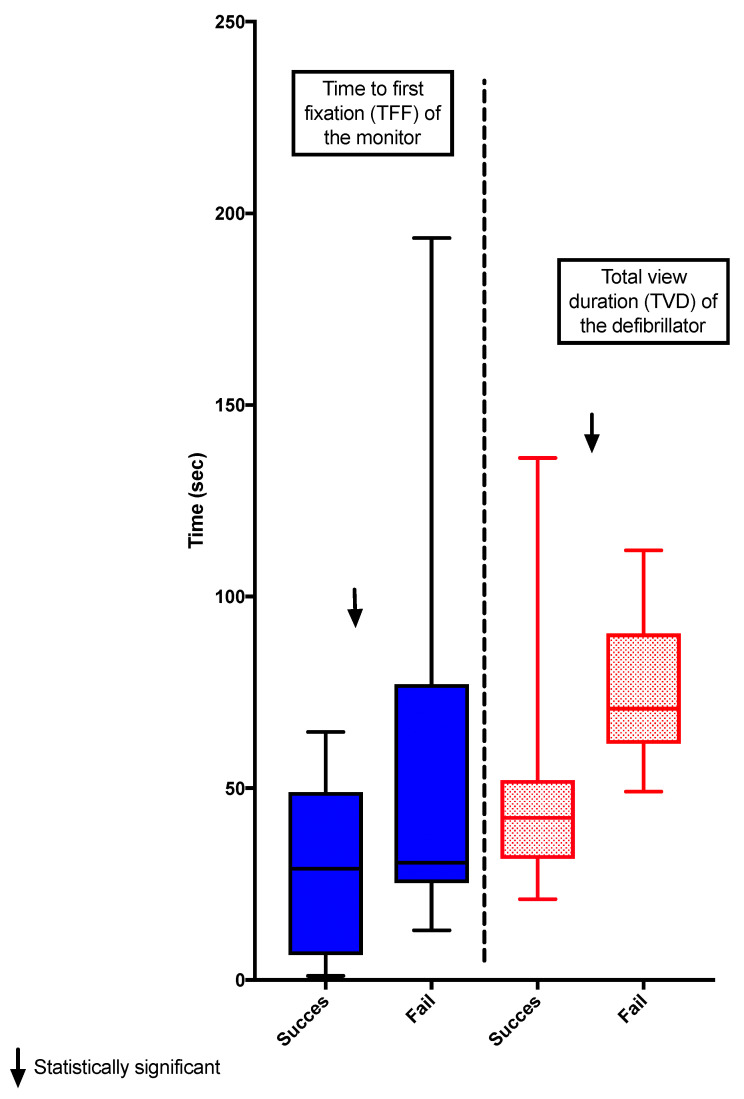
The data showed that residents who succeeded had a significantly shorter median [IQR] TVD at the pacing unit than the ones who failed (42 [31–52] sec vs. 70 [61–90] sec, *p* = 0.0097). The residents who succeeded also had a shorter median [IQR] TFF at the monitor compared with those who failed (28 [6–48] sec vs. 30 [25–77] sec, *p* = 0.5532). The boxplots show the median and IQR.

**Figure 3 behavsci-11-00031-f003:**
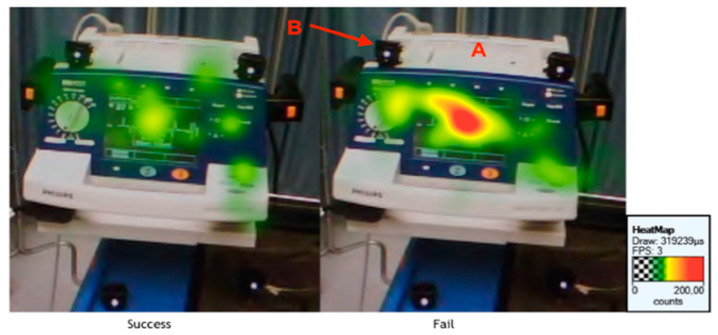
A heat map showing that the pacing unit exhibits fewer fixations (green colored) for the residents who succeeded than for those who failed (red-colored) A: Defibrillator/Pacing Unit B: Infrared markers.

**Table 1 behavsci-11-00031-t001:** Comparisons of the TVD between the three AOIs, among participants who passed the scenario and among participants who did not. Kruskal-Wallis comparison test with Dunn’s multiple comparisons test. AOI: Area of Interest.

Compared AOIs	Mean First AOI	Mean Second AOI	Mean Diff.	Adjusted *p*-Value
Participants who passed the scenario				
Monitor vs. Defibrillator (sec)	11.00	28.42	−17.42	0.0002
Monitor vs. Head of the patient (sec)	11.00	16.08	−5.08	0.7118
Defibrillator vs. Head of the patient (sec)	28.42	16.08	12.33	0.0124
Participants who did not pass the scenario				
Monitor vs. Defibrillator (sec)	5.33	15.50	−10.17	0.0029
Monitor vs. Head of the patient (sec)	5.33	7.66	−2.33	>0.9999
Defibrillator vs. Head of the patient (sec)	15.50	7.66	7.83	0.0331

## Data Availability

The data presented in this study are available on request from the corresponding author. The data are not publicly available due to privacy.

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
