# Peer review of "Comparing the Visual Perception According to the Performance Using the Eye-Tracking Technology in High-Fidelity Simulation Settings"

_behavsci, 2021, doi:10.3390/bs11030031_

Round 1
Reviewer 1 Report
The study investigates whether there are differences in fixation distribution and duration between first-year medical school residents who performed a clinical simulation successfully and those who failed. Participants had to externally pace a model of a patient (two-mannequin-model ) with unstable bradycardia. The vital signs were shown on a monitor, the task was to properly install the pacing material and stabilize the pulse in 9 minutes or less. Unless pacing was set correctly, the simulated patient continued to have nausea and being hypotensive.
Three Areas of Interest were considered: one around the monitor showing the ECG and vital signs, the second one around the head of the patient and the third one around the the defibrillator/pacing unit participants had to set up.
For each AOF, the Total View Duration (TVD) -time spent looking at an AOI during the whole simulation-, and Time to First Fixation (TFF) -the interval needed by to take the first look at the AOI, were measured.
Twelve participants managed the scenario successfully while six failed.
Participants who managed the scenario successfully looked at the ECG monitor, the only place where bradycardia was shown, earlier than those who failed. Also, they looked at the pacing unit for a shorter time than the ones who failed.
The authors conclude that results suggest a connection between the gaze pattern and performance in a high-fidelity clinical simulation scenario.
I think data support the conclusions, which are limited by worth to publish.
I only have minor comments or suggestions:
#The authors state that the heat maps depend both on number and duration of fixations, but then only write that
“Each time the eyes point to a pixel, the number for that pixel goes up by 1. As the number of fixations increases, the color on the heat map becomes « hotter »”. Here it is not clear how duration is taken into account to build the heat maps. My guess is that “each time” means each time sample from the eye-tracker. This should probably be specified.
#Figure 3. Labels are not readable+
#As the authors seem to suggest, the reported differences in TVD can be independent of the AOI, and rather reflect general differences between the two groups (e.g. the least skilled residents produced more fixations as the task for for them more difficult). This possibility could be tested with further analyses, e.g. comparing relative TVD between AOIs. This might allow expanding the conclusions.
Author Response
#The authors state that the heat maps depend both on the number and duration of fixations, but then only write that “Each time the eyes point to a pixel, the number for that pixel goes up by 1. As the number of fixations increases, the color on the heat map becomes « hotter ». Here it is not clear how duration is taken into account to build the heat maps. My guess is that “each time” means each time sample from the eye-tracker. This should probably be specified.
- Thank you for pointing out this confusing part. We have corrected the text as you rightly suggest.
#Figure 3. Labels are not readable+
- The photo quality has been improved. Thank you.
#As the authors seem to suggest, the reported differences in TVD can be independent of the AOI, and rather reflect general differences between the two groups (e.g. the least skilled residents produced more fixations as the task for them more difficult). This possibility could be tested with further analyses, e.g. comparing relative TVD between AOIs. This might allow expanding the conclusions.
- Thank you for the suggestion. We made comparisons of the TVD between the three AOIs, among participants who passed the scenario on the one hand and among participants who did not. The results showed a similar gaze distribution profile among AIOs regardless of the outcome of the scenario. All participants spent significantly more time looking at the defibrillator than at the patient's head or the monitor and spent a similar amount of time looking at the monitor and the patient's head (Kruskal-Wallis Comparison test with Dunn's multiple comparisons test). We added this part to the results section.
Reviewer 2 Report
The authors proposed the use of eye-tracker to investigate ocular behaviour of medical doctor trainees as proxy of a predictive measure of their cognitive performance. They pointed out interesting correlations between specific features (TVD, TFF) and the resulting performance, paving the way for interesting application of such technology in training protocols. Here below only some minor concerns:
- INTRODUCTION: Experimental hypothesis are based on the correlation between ocular behaviour and specific cognitive phenomena related to visual perception (e.g. workload, attention, etc.), as correctly introduced by the authors in the first lines of the manuscript, however such experimental hypothesis are not supported by evidences from literature (even being available). In order to provide to the reader a clear overview of the state of art, I encourage the authors to enlarge Introduction with a small review of recent findings supporting their hypothesis, even coming from other fields of application. For example, in https://www.frontiersin.org/articles/10.3389/fnhum.2019.00296/full it is demonstrated how the Time-To-First-Fixation of air traffic controllers during a high vigilance condition is lower with respect to a low vigilance condition, but in literature there are several more examples.
- LINE 102-103: “The first one featured the ECG and vital signs monitor and the second one the head of the patient.” The sentence is not coherent with the order of pictures in Figure1. The authors should keep the same order in order to avoid misunderstanding.
- LINE 125-126: all the information related to participants is not necessary, especially here. Only the number of actual participants is needed, as you correctly did at lines 56-58. Please there, indicate the gender balance and the age range of participants sample.
- DISCUSSION: I really liked the exhaustive discussion of the results, also including strengths and limitations of the study. I would only suggest to the authors to consider integration of eye-tracking and other forefront neuroscientific techniques, such as wearable EEG (e.g. https://www.mdpi.com/1424-8220/19/6/1365; https://jov.arvojournals.org/article.aspx?articleid=2297282; https://doi.org/10.1037/a0023885; https://www.frontiersin.org/articles/10.3389/fnins.2017.00325/full; and similars) for human factor assessment. This potential exploitation could be briefly discussed.
- Formatting check (in particular the inter-line space looks no homogeneous) and English proof-reading are encouraged.
Author Response
- INTRODUCTION: Experimental hypothesis are based on the correlation between ocular behaviour and specific cognitive phenomena related to visual perception (e.g. workload, attention, etc.), as correctly introduced by the authors in the first lines of the manuscript, however such experimental hypothesis are not supported by evidences from literature (even being available). In order to provide to the reader a clear overview of the state of art, I encourage the authors to enlarge Introduction with a small review of recent findings supporting their hypothesis, even coming from other fields of application. For example, in https://www.frontiersin.org/articles/10.3389/fnhum.2019.00296/full it is demonstrated how the Time-To-First-Fixation of air traffic controllers during a high vigilance condition is lower with respect to a low vigilance condition, but in literature there are several more examples.
- Thank you for the suggestion: the introduction has been greatly enhanced by the addition of current knowledge that can link the use of eye-tracking technology to performance, in the medicinal field and outside it (4 references).
- LINE 102-103: “The first one featured the ECG and vital signs monitor and the second one the head of the patient.” The sentence is not coherent with the order of pictures in Figure1. The authors should keep the same order in order to avoid misunderstanding.
- Thank you for the feedback.
- LINE 125-126: all the information related to participants is not necessary, especially here. Only the number of actual participants is needed, as you correctly did at lines 56-58. Please there, indicate the gender balance and the age range of participants’ sample.
- Thank you for the feedback.
- DISCUSSION: I really liked the exhaustive discussion of the results, also including strengths and limitations of the study. I would only suggest to the authors to consider integration of eye-tracking and other forefront neuroscientific techniques, such as wearable EEG (e.g. https://www.mdpi.com/1424-8220/19/6/1365; https://jov.arvojournals.org/article.aspx?articleid=2297282; https://doi.org/10.1037/a0023885; https://www.frontiersin.org/articles/10.3389/fnins.2017.00325/full; and similars) for human factor assessment. This potential exploitation could be briefly discussed.
- Thank you for the suggestion. The discussion was indeed enriched by a brief paragraph on the value of adding the EEG to the eye-tracking technology.
- Formatting check (in particular the inter-line space looks no homogeneous) and English proof-reading are encouraged.
- Thank you for the suggestion “English proof-reading has been performed by a native English co-author (PD).
Reviewer 3 Report
The paper "Examining the Visual Perception and the Performance Using the Eye-Tracking Technology" is about a studie high-fidelity scenario involving a patient 16 in shock using eye tracking. They recorded eighteen participants and tried to find statistical gaze information which holds the information of success or failure.
Positive:
- Good description of the studie
- Very good limitations section.
My main criticisms on this work concerns the small number of subjects involved in the study, especially the fact that there is only one age group, and the sparse evaluation. The authors have mentioned some of these limitations themselves in the limitation section, but this does not change the fact that it not sufficient for publication in its current form. Personally, I find the topic highly interesting and it is also important for reviewing the skills of clinicians. I recommend the authors to increase the number of subjects and to extend the evaluations.
Possible improvements:
- Add a scanpath based classification
- Analyse the AOI transitions
- Evaluate more features statistically (as Table or Plot)
- Use the 2 degree of the fovea to produce the heatmap
Minor:
- Titel is too general
- Figure 3 has low quality
Author Response
- My main criticisms on this work concerns the small number of subjects involved in the study, especially the fact that there is only one age group, and the sparse evaluation. The authors have mentioned some of these limitations themselves in the limitation section, but this does not change the fact that it not sufficient for publication in its current form.
- Thank you for your suggestion. We fully understand the limitation related to the small sample size of this study. Unfortunately, we cannot add more participants as the study is currently completed, but we have assigned other analyzes to quantify the magnitude of the results found. Here is our response to the editor who raises the same limitation.
- Response to the editor: Thank you for this comment. We are fully aware of this limitation and have specifically added it in the limitation section of the article. As the study is currently completed, we can no longer increase the number of participants recruited. Our observational study will undoubtedly, as a seed study, serve to calculate the sample size of other similar ones. As we noted in the study limitations section, the convenience sample chosen was not previously calculated. The similar studies previously carried out, are mainly also exploratory, with an objective often different from ours or in other fields other than health and patient safety. These studies do not allow sample size estimation.
To assess, in retrospect, whether our sample size was adequate, using the t value and the degrees-of-freedom (df) of the unpaired t-test, we calculated the effect size (r) to characterize the magnitude of the difference between the groups. An r value of 0.2 was considered a “weak” effect, 0.5 “medium,” and 0.8 “strong”. (Becker L. Calculate d and r using t values and df (separate groups t-test): University of Colorado; 2000, Available from https://lbecker.uccs.edu/ ) (This part was added in the statistics section of the manuscript). The effect size r was respectively 0.44 and 0.45 for the TVD of the defibrillator and the TFF of the monitor. This indicates a medium difference between the groups, although it was significant, and might suggest that the sample size should be larger (This part was added in the statistics section of the manuscript).
- Title is too general: Corrected. Thank you for the suggestion: Comparing the Visual Perception According to the Performance Using the Eye-Tracking Technology in a High-Fidelity Simulation Settings.
- Figure 3 has low quality: Corrected. Thank you.
Round 2
Reviewer 2 Report
The authors carefully took into consideration and addressed all the arisen concerns, the manuscript has been improved thus it is now suitable for publication.
Author Response
The manuscript references will be reinforced based on reviewer 3 comments. Thank you for your greatly helpful feedback.
Reviewer 3 Report
This manuscript looks into gaze as gaze as a relation to successful performance in a clinical procedure brings little novel contribution to research in eye movements and medical decision making.
The most outstanding concern is that there is no concrete hypothesis. A hypothesis that simply sets out to see if there are differences in the gaze behavior of successful and unsuccessful performance lacks in contrast to the actual results the authors report. Hypothesizing that you expect to see differences in the gaze behavior has already been found in countless literature on eye movements in medical experts, novices, and residents. The authors have even pointed this out in multiple paragraphs of the introduction and the discussion. The works cited in this manuscript have much more to offer regarding how the behavior differs for gaze dispersion, AOI attention, fixation behavior and much more, but the authors neglect to go into details, which could help frame their research question. As written, the authors’ current contribution to this research is not properly highlighted right from the beginning. It comes across that whoever wrote this manuscript has little experience writing for academic journals. Even though they have put a nice effort into the experiment itself, their support and interpretation of this work is severely lacking.
-Introduction:
- The cited works seem like a stream of consciousness and barely build off one another but rather say the same thing repeatedly. Content wise the mentioned literature does not go into anything concrete regarding specific eye movements or attention to information relevant to the diagnostic decision making. In general, pulling generic statements from the literature does not set the stage for promising work and especially when there is already so much insight in the medical domain (using sources from aviation seems irrelevant for your work).
Some sources that would be key to your work should go into detail about how medical experts and even residents are more apt to filter out “irrelevant” details, which leads to faster and more accurate diagnoses. These cognitive processes are evident with more fixations in relevant AOIs and shorter time to first fixation to these AOIs.
- A. Gegenfurtner, E. Lehtinen, and R. Säljö, “Expertise differences in the comprehension of visualizations: A meta-analysis of eye-tracking research in professional domains,” Educational Psychology Review, vol. 23, no. 4, pp. 523–552, 2011.
- A Van der Gijp, C. Ravesloot, H Jarodzka, M. van der Schaaf, I. van der Schaaf, J. P. van Schaik, and T. J. Ten Cate, “How visual search relates to visual diagnostic performance: A narrative systematic review of eye-tracking research in radiology,” Advances in Health Sciences Education, vol. 22, no. 3, pp. 765–787, 2017
- N. Castner, T. Appel, T. Eder, J. Richter, K. Scheiter, C. Keutel, F. Hüttig, A. Duchowski, and E. Kasneci, “Pupil diameter differentiates expertise in dental radiography visual search,” Plos One, vol. 15, no. 5, e0223941,2020.
T. Tien, P. H. Pucher, M. H. Sodergren, K. Sriskandarajah, G.-Z. Yang, and A. Darzi, “Differences in gaze behaviour of expert and junior surgeons performing open inguinal hernia repair,” Surgical Endoscopy, vol. 29, no. 2, pp. 405–413, 2015.
H. Haider and P. A. Frensch, “Eye movement during skill acquisition: More evidence for the information-reduction hypothesis.,” Journal of Experimental Psychology: Learning, Memory, and Cognition, vol. 25, no. 1, p. 172, 1999.
Eivazi, A. Hafez, W. Fuhl, H. Afkari, E. Kasneci, M. Lehecka, and R. Bednarik, “Optimal eye movement strategies: A comparison of neurosurgeons gaze patterns when using a surgical microscope,” Acta Neurochirurgica, vol. 159, no. 6, pp. 959–966, 2017.
- As of now, the work cited in your intro does little to help you. Additionally, reference 38 appears to be research that is very similar to this current work. This should be referenced in the introduction. How are you building off their work and what are their findings? How will your findings differ in order to further understand eye movements of successful performance in high fidelity situations? This is where your hypothesis should be ingrained and have a specific course of what the reader expects to see from your results and be able to understand that this work makes a significant contribution.
-Methods:
- This is the redeeming part of your manuscript, but there are significant issues with the structure and understandability.
- Redundancies. For example, in the participant information, lines 160 and 162 you mention first year residents twice. There is also participant information in the results section (lines 334 to 342) that is better suited for this section and/or is unnecessarily repeated.
- The sentence starting at line 162 is not a sentence.
- The calibration information needs more information, so readers get a clear picture. How many points calibration? How far away was the participant standing from the calibration window or sign? Is this similar to how far away the participant was to the stimuli. line 217 is unclear. Did they have the prescription lenses only on for the calibration? Or the whole time? Is 75% a good calibration threshold and how did you pick this value. Was there any calibration after the experiment to account for gaze signal inaccuracies due to slippage?
- In general, were the participants allowed to move around, were they instructed to standstill, are they standing or sitting at monitor? It is very hard to follow what the environment for the data collection was like.
- The scenario. For someone who has no understanding of this medical procedure (as this may be the case for the majority of readers of this journal). The scenario is very technically wordy and hard to grasp what is actually going on and important to a participant’s success. I suggest condensing and keep some of the technical terms but have some short descriptions. Sentences at 225 and 239 are particularly hard to follow due to long winded and convoluted wording.
- Data Collection: I’m curious if the squared IR AOIs become distorted if there was a perspective change, and if distortions affect the recognition of a fixation in that AOI? Sentence at 309 sounds better as heatmaps relative to each AOI are extracted.
- Eye trajectories sounds like eyeballs flying through space. “eye movements” or “gaze trajectories”
- Heat map definition is too long and confusing that it sounds incorrect. Shorten to one sentence and base it off a definition from Andrew Duchowski or Kenneth Holmqvist and be sure to cite them.
- Statistical methods. Why did you choose the Mann-Whitney over a t-test? Please provide good reasoning. And what software did you use? SPSS or R and which version.
-Results:
- line 347 on heatmaps redundant
- sentence structuring hard to read sometimes. “for variable, we found blank”. Can sometimes make the subject of the sentence clearer to the reader.
- line 374 is more appropriate for the limitations part of the discussion.
- line 376, 1 sentence paragraph. And a minor suggestion could be use milliseconds instead of seconds, but it is your preference.
- line 378 use of on the one hand is sloppy.
- line 380 AIOs
- the last paragraph you write that there were significant findings, but you do not report these values. If the values are not reported no one will believe you.
-Figures:
- make sure figure and caption are on the same page!
- because of the placement of figure 2 and 3, you have half a blank page. Please consider repositioning so there is no large whitespace.
- figure 2. Are the boxplots showing the median and IQR since these are the values you are reporting in your results? This should be clear in your caption.
- figure 3. Not shorter, less fixations.
-Discussion:
- way too long and redundant. It should be a summary of your findings and how they can be interpreted in the scope of how your work contributes to the research in this field.
- poor writing in sentence 457
- line 460 is way too passive, which undermines all the work you put into running this experiment.
- line 462 is a gross overstatement and leads me to think you have not properly researched.
- line 465 to the end of the key results goes completely downhill. You are not supporting your findings, but back to restating that there are differences in the gaze behavior, which we know already. You also invalidate all your work again in lines 487-489 and later again in your conclusion (lines 612-613).
- There is no need to repeat what you said in the intro. And it is very sloppy to actually repeat sentences, e.g., at lines 465 and 491, 468 and 494, 472 and 499.
- be careful with your limitations, you make them sound like excuses when they are definitely valid limitations.
Author Response
Reviewer 3 : Thank you very much for your very thoughtful feedback.
- The most outstanding concern is that there is no concrete hypothesis : Thank you for your comment. We agree that the hypothesis is not elaborate enough, for fear of being "excessively" speculative, for an exploratory study. But, as you well suggest, we have deepened the rationale behind the hypothesis based on the literature you have brought to our attention.
-Introduction:
- The cited works seem like a stream of consciousness and barely build off one another but rather say the same thing repeatedly. Content wise the mentioned literature does not go into anything concrete regarding specific eye movements or attention to information relevant to the diagnostic decision making. In general, pulling generic statements from the literature does not set the stage for promising work and especially when there is already so much insight in the medical domain (using sources from aviation seems irrelevant for your work).
Some sources that would be key to your work should go into detail about how medical experts and even residents are more apt to filter out “irrelevant” details, which leads to faster and more accurate diagnoses. These cognitive processes are evident with more fixations in relevant AOIs and shorter time to first fixation to these AOIs.
- As of now, the work cited in your intro does little to help you. Additionally, reference 38 appears to be research that is very similar to this current work. This should be referenced in the introduction. How are you building off their work and what are their findings? How will your findings differ in order to further understand eye movements of successful performance in high fidelity situations? This is where your hypothesis should be ingrained and have a specific course of what the reader expects to see from your results and be able to understand that this work makes a significant contribution.
Thank you for the references that we have included in the introduction. We also substantiated the introduction and explained more specifically our hypothesis and the rationale behind our hypothesis. Finally, as suggested, we explained what sets current research apart from what is already published. Thank you for the suggestion of a thorough review of the introduction.
-Methods:
- Redundancies. For example, in the participant information, lines 160 and 162 you mention first year residents twice. There is also participant information in the results section (lines 334 to 342) that is better suited for this section and/or is unnecessarily repeated. : Thank you for underlining this repetition. The correction has been made.
- The sentence starting at line 162 is not a sentence. : Thank you for your comment. Corrected.
- The calibration information needs more information, so readers get a clear picture. How many points calibration? How far away was the participant standing from the calibration window or sign? Is this similar to how far away the participant was to the stimuli. line 217 is unclear. Did they have the prescription lenses only on for the calibration? Or the whole time? Is 75% a good calibration threshold and how did you pick this value. Was there any calibration after the experiment to account for gaze signal inaccuracies due to slippage? In general, were the participants allowed to move around, were they instructed to standstill, are they standing or sitting at monitor? It is very hard to follow what the environment for the data collection was like. Data Collection: I’m curious if the squared IR AOIs become distorted if there was a perspective change, and if distortions affect the recognition of a fixation in that AOI? : Thank you for the comment. Details about the calibration technique and the settings were added in the method section.
“ Glasses were fit to each resident’s face and calibrated following the manufacturer’s instructions. Prescription lenses were kept for calibration and during the experiment. During the scenario, the participant was allowed to move around and take care of the simulated patient as if he was in real clinical practice. In a time-wise fashion, we performed 3 calibration attempts to reach at least 75% accuracy and tracking scores, as it was recommended by the manufacturer. To do so, the participant stood one meter away from a whiteboard (a square of 1 meter side) on which four infrared sensors were hung. After wearing the glasses (over his prescription lenses if necessary), the participant had to follow, only with his eyes, without moving his head, a 5th central sensor that the investigator made move following a z-shape from bottom to top of the boar, then from right to left. No calibration was needed after the experiment. Once calibrated the squared IR AOIs can become distorted with peripheral vision, but the distortions do not affect the recognition of a fixation in that AOI.”
- Sentence at 309 sounds better as heatmaps relative to each AOI are extracted. : Corrected. Thank you.
- The scenario. For someone who has no understanding of this medical procedure (as this may be the case for the majority of readers of this journal). The scenario is very technically wordy and hard to grasp what is actually going on and important to a participant’s success. I suggest condensing and keep some of the technical terms but have some short descriptions. Sentences at 225 and 239 are particularly hard to follow due to long winded and convoluted wording. : Thank you for the suggestion. Explanations and simplification of the medical terms have been added in the scenario section.
- Eye trajectories sound like eyeballs flying through space. “eye movements” or “gaze trajectories” : Corrected
- Heat map definition is too long and confusing that it sounds incorrect. Shorten to one sentence and base it off a definition from Andrew Duchowski or Kenneth Holmqvist and be sure to cite them. Corrected. Thank you for the suggestion and the references.
- Statistical methods. Why did you choose the Mann-Whitney over a t-test? Please provide good reasoning. And what software did you use? SPSS or R and which version. : Thank you for your comment. We have added “ Because of the limited sample size and the non-normal distribution of the data (D’Agostino & Pearson normality test), the Mann-Whitney U test was used …” and also “Statistical analysis was performed using Prism 9.0.1 for Mac OS X (GraphPad Software, La Jolla, CA).”.
-Results:
- line 347 on heatmaps redundant : Corrected. Thank you.
- sentence structuring hard to read sometimes. “for variable, we found blank”. Can sometimes make the subject of the sentence clearer to the reader.: Corrected. Thank you.
- line 374 is more appropriate for the limitations part of the discussion. : Thank you for the comment. This sentence has been deleted. It is repeated in the limitations section.
- line 376, 1 sentence paragraph. : This part has been split into two sentences. Thank you for the suggestion.
- line 378 use of on the one hand is sloppy. : Corrected. Thank you.
- line 380 AIOs : Corrected. Thank you.
- the last paragraph you write that there were significant findings, but you do not report these values. If the values are not reported no one will believe you. : Results were added in table 1 as suggested. Thank you.
-Figures:
- make sure figure and caption are on the same page! : We edited the manuscript so that the figures and captions are now on the same page. It is still possible that they appear to be not on the same page on the tracked modifications version of the manuscript, but we have made sure that they are in the clean version.
- because of the placement of figure 2 and 3, you have half a blank page. Please consider repositioning so there is no large whitespace. : Corrected. Thank you. Table 1 currently fills this space (see clean version).
- figure 2. Are the boxplots showing the median and IQR since these are the values you are reporting in your results? This should be clear in your caption. : Thank you. In the caption of the figure, you can read “The data showed that residents who succeeded had a significantly shorter median [IQR] TVD at the pacing unit than the ones who failed (42 [31-52] sec vs 70 [61-90] sec, p=0,0097). The residents who succeeded also had a shorter median [IQR] TFF at the monitor compared with those who failed (28 [6-48] sec vs 30 [25-77] sec, p=0,5532). The boxplots show the median and IQR. »
- figure 3. Not shorter, less fixations. : Corrected. Thank you. “fewer fixations”
-Discussion:
- way too long and redundant. It should be a summary of your findings and how they can be interpreted in the scope of how your work contributes to the research in this field. Thank you for the suggestion. We reorganized the discussion, removed some repetitive sentences (as These results suggest that there might be a connection between the gaze pattern of residents in a high-fidelity simulation and their performance), emphasized the difference between the current literature and our research. We kept the reference related to the field of aviation as requested by another reviewer.
- poor writing in sentence 457 : Corrected. Thank you
- line 460 is way too passive, which undermines all the work you put into running this experiment. : Corrected. Thank you.
- line 462 is a gross overstatement and leads me to think you have not properly researched. : This sentence has been deleted. Thanks for the feedback
- line 465 to the end of the key results goes completely downhill. You are not supporting your findings, but back to restating that there are differences in the gaze behavior, which we know already. You also invalidate all your work again in lines 487-489 and later again in your conclusion (lines 612-613). : These sentences were rewritten or removed to give more merit to our study and not to contradict the elements demonstrated in our research.
- There is no need to repeat what you said in the intro. And it is very sloppy to actually repeat sentences, e.g., at lines 465 and 491, 468 and 494, 472 and 499. : Corrected. Thank you.
- be careful with your limitations, you make them sound like excuses when they are definitely valid limitations. : The limitation section has been edited to emphasize the limits of our methodological choices (without justifying them). Thanks for the suggestion.